# Intravenous Ferric Carboxymaltose for the Treatment of Iron Deficiency Anemia During Pregnancy: Effects on Maternal and Fetal Wellbeing—A Multicenter Retrospective Observational Study

**DOI:** 10.3390/nu17162670

**Published:** 2025-08-19

**Authors:** Eleonora Romani, Sara Zullino, Anna R. Speciale, Paola M. Villa, Veronica Bonaldo, Francesca Parisi, Chiara Lubrano, Felice Petraglia, Irene Cetin, Federico Mecacci

**Affiliations:** 1Obstetrics and Gynaecology Unit, Department of Experimental and Clinical Biomedical Sciences, University of Florence, 50134 Florence, Italy; eleonora.romani@unifi.it (E.R.); annarosa.speciale@unifi.it (A.R.S.); felice.petraglia@unifi.it (F.P.); federico.mecacci@unifi.it (F.M.); 2High Risk Pregnancy Unit, Department for Women and Health, Careggi University Hospital, 50134 Florence, Italy; sarazullino@hotmail.it; 3Unit of Obstetrics and Gynaecology, Department of Mother, Woman and Neonate, Hospital Vittore Buzzi, 20154 Milan, Italy; paola.villa@asst-fbf-sacco.it (P.M.V.); veronica.bonaldo@unimi.it (V.B.); 4S.C. Ostetricia, Fondazione IRCCS Cà Granda, Ospedale Maggiore Policlinico, 20122 Milan, Italy; francesca.parisi@unimi.it (F.P.); chiara.lubrano@unimi.it (C.L.); 5Department of Biomedical and Clinical Sciences, University of Milan, 20157 Milan, Italy; 6Nutritional Sciences—Doctoral Programme, Department of Veterinary Medicine and Animal Sciences (DIVAS), University of Milan, 26900 Lodi, Italy; 7Department of Clinical and Community Sciences, University of Milan, 20122 Milan, Italy

**Keywords:** iron deficiency anemia, ferric carboxymaltose, cardiotocography, pregnancy

## Abstract

**Objectives**: To assess the impact of intravenous ferric carboxymaltose (FCM) on fetal and maternal adverse effects in pregnant women diagnosed with iron deficiency anemia (IDA). **Methods**: This is a multicenter retrospective study on 472 pregnant women diagnosed with moderate to severe IDA undergoing treatment with FCM between 2019 and 2025 at Careggi University Hospital (Florence) and Vittore Buzzi Children Hospital (Milan). Fetal wellbeing was assessed using computerized cardiotocography (cCTG) or ultrasound, based on gestational age at treatment. Maternal side effects were evaluated through clinical evaluation. **Results**: cCTG was performed in 377/472 patients (80%), with a mean short-term variability of 10.2 ms. Normal cCTG criteria were met in 98.4% of cases; six patients exhibited transient reduced variability, which resolved following intrauterine resuscitation. Ultrasound assessment was performed in 95 patients (20%), revealing no fetal heart rate abnormalities. Maternal side effects occurred in seven patients (1.4%). Hemoglobin concentrations increased by a median of 1.4 g/dL after five weeks, reaching up to 2.8 g/dL in women with a baseline Hb < 8 g/dL. **Conclusions**: Our findings support the potential safety and efficacy of intravenous FCM for the treatment of IDA during pregnancy, demonstrating low rates of maternal side effects and no adverse fetal heart rate patterns. However, prospective studies are necessary to confirm these results.

## 1. Introduction

Anemia in pregnancy is a global problem affecting 36.5% of women [1]. It is defined as hemoglobin (Hb) levels below 11 g/dL in the first and third trimesters, below 10.5 g/dL in the second trimester [2,3] and less than 10 g/dL in the postpartum period [4]. It is categorized by the World Health Organization as severe (Hb < 7 g/dL), moderate (7–9.9 g/dL) and mild (10–10.9 g/dL) anemia [5]. Globally, iron deficiency (ID) accounts for over 60% of anemia diagnoses [6] and the pooled global prevalence of iron deficiency anemia (IDA) among pregnant women is estimated to be approximately 19% [7].

Pregnancy predisposes to ID and IDA due to increased demands (ranging from 0.8 to 7.5 mg of iron/day from the first to the third trimester) required to support maternal plasma volume expansion and red blood cell (RBC) production, fetal–placental growth and to compensate for blood loss at delivery [8]. Moreover, approximately 50% of women in the first trimester already exhibit low or absent iron stores, with ferritin levels below 30 ng/mL, which are insufficient to meet the high requirements of pregnancy [9]. If untreated, IDA is notoriously associated with fetal and maternal adverse outcomes, including increased risk of preeclampsia, preterm birth, fetal growth restriction, low birth weight and impaired neurodevelopment and cognitive performance in the offspring [10].

To reduce these risks, international guidelines recommend intravenous (IV) iron therapy in cases of moderate to severe anemia, intolerance or lack of response to oral treatment and hemoglobin levels below 10 g/dL at gestational weeks close to delivery (>34 weeks) [4]. IV iron has been shown to be more effective and better tolerated than oral formulations [11], with ferric carboxymaltose (FCM) emerging as the preferred agent for use during pregnancy [12]. Recent randomized controlled trials (RCTs) have demonstrated the effectiveness of IV FCM in pregnant women during the second and third trimesters [2,13]. Nevertheless, the necessity of fetal monitoring during intravenous (IV) iron administration remains a topic of debate. Recent hematology guidelines advise against routine fetal heart rate monitoring, citing the low incidence of fetal adverse events observed in clinical trials and real-world data, as well as the absence of a known mechanism linking FCM to changes in fetal heart rate [14]. Conversely, case reports have described episodes of fetal bradycardia and neonatal acidosis following IV iron in otherwise uncomplicated pregnancies [15]. Although these reports are concerning, they are not supported by large-scale evidence. This discrepancy underscores the need for further objective assessment of fetal wellbeing during FCM treatment, particularly through the use of standardized tools such as computerized cardiotocography (cCTG). Therefore, the primary aim of this study was to assess fetal wellbeing through ultrasound and cCTG parameters during IV iron administration in a large cohort of pregnant women diagnosed with IDA. Secondly, this study aimed to evaluate short-term maternal side effects and treatment effectiveness.

## 2. Materials and Methods

This is a multicenter retrospective study conducted on pregnant women diagnosed with IDA at Careggi University Hospital in Florence and at Vittore Buzzi Children’s Hospital in Milan, from 2019 to 2025. All patients provided a written informed consent prior to IV iron therapy, which also included permission for the use of their anonymized clinical data for research purposes. As the present study involved a retrospective analysis of fully anonymized, non-identifiable data, formal approval by the Institutional Review Board was not required, in accordance with the internal policies of the participating hospitals.

### 2.1. Study Population

Eligible patients were women aged at least 18 years, with moderate to severe IDA, defined as Hb ≤ 9.9 g/dL and ferritin < 30 ng/mL, diagnosed in the second or third trimester of pregnancy. Clinical indications for IV iron treatment included: (1) intolerance and/or insufficient response to oral iron supplements and/or (2) the need for rapid repletion of iron stores and/or (3) anemia symptoms (e.g., tiredness, shortness of breath or heart palpitations).

Exclusion criteria included: (1) anemia due to hematologic causes other than ID (e.g., thalassemia, hemolytic anemia or anemia of chronic disease); (2) comorbidities contraindicating IV iron therapy (e.g., active infections, severe liver disease or renal failure); (3) known hypersensitivity to FCM or any of its components; (4) first trimester of pregnancy; and (5) missing outcome data.

### 2.2. Treatment Protocol

All patients received FCM at a dose of 500 mg, diluted in 250 mL of normal saline, administered over 15 min. Each dose did not exceed 20 mg/kg of body weight, and the maximum cumulative weekly dose was 1000 mg, typically divided into one or two infusions per week. Oral iron supplementation was discontinued during the week following the intravenous infusion to optimize absorption. No dietary modifications or additional nutritional interventions were incorporated into the treatment protocol.

### 2.3. Maternal Data Collection

Maternal characteristics, comorbidities and any oral iron supplementation consumption were recorded. For each patient, Hb and ferritin levels before iron infusion, as well as Hb at the time of delivery and improvement of symptoms, were collected to evaluate the response to treatment.

Immediate maternal adverse effects were assessed by clinical examination. Patients were closely monitored for any adverse side effects during the entire infusion period and almost 30 min after the infusion.

### 2.4. Fetal Heart Rate Data

Fetal wellbeing was assessed as follows according to gestational age:-Gestational age beyond 28 weeks: computerized cardiotocography analysis (cCTG) was performed to reduce intra-operator variability in interpretation, using Short-Term Variability (STV) as an objective parameter.-Gestational age below 28 weeks: fetal wellbeing was assessed by an ultrasound (US) examination of fetal heart rate (FHR).

### 2.5. Statistical Analyses

Continuous variables were expressed as means ± standard deviations (SDs) or medians and ranges, as appropriate, and categorical variables were expressed as absolute and relative frequencies. Continuous data were tested for normality using the Shapiro–Wilk test. As Hb and ferritin values were not normally distributed (*p* < 0.05, Shapiro–Wilk test), non-parametric tests were used for comparisons.

A non-parametric Wilcoxon signed-rank test was used to compare hemoglobin (Hb) levels before and after IV iron therapy. Subsequently, a subgroup analysis was performed, stratifying patients according to pre-treatment anemia severity (Hb < 8 g/dL vs. ≥8 g/dL). The Hb increase after treatment was compared between the two subgroups using the non-parametric Mann–Whitney U test.

All statistical tests were two-sided and a *p*-value < 0.05 was considered statistically significant. All analyses were performed using SPSS software (version 30.0.0.0, IBM Corp., Armonk, NY, USA).

## 3. Results

A total of 487 pregnancies diagnosed with moderate to severe anemia and treated with FCM during the study period were evaluated. After the exclusion of 15 patients with incomplete data (missing cCTG data, pre- or post-treatment Hb values or delivery data), the total study population included 472 pregnancies, of which 53 women had a pre-treatment Hb lower than 8 g/dL.

### 3.1. Fetal Heart Rate Data

Fetal wellbeing was assessed by cCTG analysis in 377/472 (80%) cases. In 37 out of 377 (9.8%), cCTG was performed during IV administration, in 255/377 (67.6%) within the first 30 min, in 61/377 (16%) between 30 and 60 min after therapy and in 24/377 (6.4%) after 120 min. Regardless of the timing of monitoring initiation, in 371/377 (98.4%) patients the cCTG analysis met Dawes and Redman criteria, and the mean STV value was 10.2 ms. In the six cases where the criteria were not fully met, this was due to a lack of high variability episodes, which were resolved by intrauterine resuscitation interventions such as maternal IV hydration. No episodes of fetal deceleration, tachycardia or bradycardia were recorded. In 95/472 (20%) patients, fetal wellbeing was assessed by US examination immediately after IV administration and no cases of FHR abnormalities were observed (Table 1).

### 3.2. Maternal Data

Out of 472 included cases, 67 (14%) had pre-existing comorbidities predisposing to ID. Among these, 32/67 had gastrointestinal disorders leading to malabsorption (e.g., inflammatory bowel diseases, celiac disease or chronic gastritis), previous bariatric surgery or bowel resection. The remaining 35/67 patients had gynecological conditions responsible for heavy menstrual bleeding (HMB) (e.g., endometriosis, adenomyosis or uterine fibroids). Table 2 shows maternal and pregnancy characteristics of the total study population and subgroups according to pre-treatment Hb values.

A total of 308/472 (65%) patients underwent oral iron supplementation during pregnancy (dosage ranging from 27 to 200 mg/day) before FCM infusion, without anemia resolution.

The median hemoglobin level before IV iron treatment was 9.2 g/dL (IQR: 8.6–9.7) and median ferritin was 13.7 ng/mL (IQR: 7.9–16.7). The median gestational age at FCM administration was 34 weeks. At delivery, the median Hb level was 10.6 g/dL (IQR: 9.9–11.3), reflecting a median increase of 1.4 g/dL (IQR: 0.7–2.2) over a period of 5 weeks, considering that the median gestational age at birth was 39 weeks (Figure 1).

A subgroup analysis was subsequently performed, stratifying the total study population according to the severity of pre-treatment anemia (Hb < 8 g/dL versus ≥8 g/dL). A total of 53/472 (11.2%) women had an Hb < 8 g/dL and 419/427 (88.7%) had values > 8 g/dL. The median Hb increase after treatment was significantly higher in the group with more severe anemia (2.8 [1.7–3.6] g/dL) compared to the group with milder anemia (1.3 [0.6–2.0] g/dL, *p* < 0.001, Mann–Whitney U test) (Table 3).

A total of 190 out of 472 patients (40%) achieved resolution of anemia, reaching Hb levels above 11 g/dL at term, with an average of 1.6 administered doses. A total of 3 cases out of 472 (0.6%) had Hb concentrations below 8 g/dL at the time of delivery despite IV treatment. Maternal side effects were recorded in 7/472 (1.4%) cases and they were primarily represented by episodes of nausea, vomiting and hypotension.

All successfully treated patients reported an improvement in anemia symptoms at the time of delivery, particularly fatigue.

## 4. Discussion

The present retrospective study investigates the safety and effectiveness of IV iron therapy among pregnant women diagnosed with moderate to severe IDA. Our results showed a reassuring safety profile for the fetus, with no adverse events or cardiac rhythm alterations as detected by cardiotocography and ultrasound assessment. However, the retrospective design limits the ability to draw causal inferences.

Although IV iron is generally considered safe and effective for treating maternal iron deficiency anemia, rare reports of fetal distress, including bradycardia and acidosis, following its administration have been described [15]. The exact mechanisms are not fully understood but may involve transient alterations in placental blood flow or maternal–fetal oxygen exchange.

Data from two pharmacovigilance databases reinforced the evidence that iron isomaltoside 1000 is associated with a higher incidence of severe hypersensitivity reactions compared to FCM [16]. In 2015, Woodward et al. documented a case involving a woman who required an emergency cesarean section following an intravenous infusion of iron isomaltoside 1000 [17]. Conversely, Froessler et al. reported a favorable safety profile of FCM in a cohort of 863 pregnant women, showing minimal adverse effects and no significant risks to the fetus or mother when administered appropriately [18].

In July 2019, the European Medicines Agency (EMA) issued a statement indicating that fetal bradycardia may occur following the administration of parenteral iron. This condition is typically transient and results from a hypersensitivity reaction in the mother. The agency emphasized the importance of careful fetal monitoring during intravenous iron administration in pregnant women. This recommendation has been communicated to the Italian Medicines Agency (AIFA) and incorporated into the summary of product characteristics for all iron formulations. Nevertheless, our results showed that 98.4% of the present cohort met the cCTG criteria analysis and no episodes of abnormal FHR were recorded, even when the evaluation was carried out within 60–120 min. No emergency delivery was performed during or after IV iron infusion. This finding aligns with recent studies that did not report significant adverse fetal outcomes following IV iron administration [18].

Anemia in pregnancy is mostly related to a physiologic process of hemodilution, caused by a higher expansion of plasma volume compared to RBC production (50% versus 15–25%), in order to facilitate nutrient and oxygen delivery to the fetus [19]. Consequently, maternal iron demands significantly increase during pregnancy to support the hematopoiesis process. Therefore, ID is the most common nutritional deficiency in pregnancy due to an imbalance between demand and supply. As a recent retrospective Canadian study showed, half of women already have insufficient iron stores during the first trimester of pregnancy and 8% are anemic [9], mostly due to a diet low in iron-rich foods or concomitant comorbidities.

In our cohort, 67 out of 472 patients (14%) had pre-pregnancy conditions leading to ID, including about 50% of women affected by adenomyosis, endometriosis or uterine fibroids responsible for HMB. Actually, HMB is recognized as the most common cause of IDA in women of reproductive age, as every 30–50 mL of menstrual blood loss results in a depletion of 25–50 mg of iron per cycle [20].

Regarding IDA management, oral ferrous iron salts are first line treatment options [2,3,4,11], with Hb checked again after 2–3 weeks to assess compliance, correct administration and response to treatment [19,21].

In our cohort of patients, 308/472 (65%) had taken a daily oral supplementation during pregnancy without anemia resolution. Potential reasons for a lack of response include non-adherence to therapy, anemia related to causes other than ID and reduced absorption due to co-existing comorbidities [19,21]. According to this, 32/67 (47.7%) women in our cohort had gastrointestinal disorders, such as inflammatory bowel diseases, celiac disease, chronic gastritis, or previous bariatric surgery or bowel resections.

Another cause of poor response to therapy may be partially related to the mechanism of oral iron absorption itself. Oral supplements cause an increase in serum hepcidin levels for approximately 24 h, which is sufficient to impair hepatocyte ferroportin and reduce iron gut absorption from subsequent doses [22]. Based on this mechanism and to enhance iron absorption, recent data proposed an alternate-day dosing regimen instead of a daily one and, in cases of insufficient response, IV administration [22]. In fact, unlike oral formulations, IV iron is absorbed rapidly by macrophages and released into the circulation to be bound to transferrin. Since macrophage ferroportin expression is much higher than that in the gut, a higher amount of hepcidin is required to effectively inhibit it [23]. This makes IV administration more effective than oral iron and is associated with fewer gastrointestinal side effects.

In our cohort of patients, there was a median increase in Hb levels of 1.4 g/dL after treatment over a period of approximately 5 weeks. This improvement is clinically meaningful, particularly considering that 40% of patients achieved Hb levels over 11 g/dL at delivery. Regardless of the average increase in Hb level after therapy, which can be influenced by several factors (e.g., patient comorbidities, anemia severity, number of doses administered and trimester of pregnancy), there is substantial agreement that FCM is associated with higher Hb levels after therapy compared to other IV iron or oral formulations [11].

In our study, patients with severe IDA had a higher response to treatment compared to those with moderate anemia, with a median Hb increase of 2.8 g/dL. This increase is lower than the recently reported value of 4.23 g/dL in an Indian retrospective study, which was conducted on a very small sample of patients [24].

Only 3 women out of 472 (0.6%) had an Hb < 8 g/dL at the time of delivery despite IV iron treatment. This result may be explained by the timing of the iron infusion, which in all cases occurred less than two weeks before delivery. In fact, the expected response to iron treatment is normally represented by an initial reticulocytosis after approximately one week, followed by an increase in Hb of at least 1 g/dL within two to four weeks after treatment [19,21].

Regarding maternal safety of FCM therapy in our cohort, nausea, hypotension and vomiting were the most frequent adverse effects, recorded in only 0.4% of cases, due to FCM’s low immunogenic potential and molecular stability. Our cohort reported a reduction in anemia related symptoms following the observed increase in Hb levels and anemia resolution.

## 5. Future Research

Although our study focused exclusively on FCM, emerging evidence has raised concerns regarding its association with hypophosphatemia, a condition that may exacerbate fatigue and other anemia symptoms. This adverse effect has been observed more frequently with FCM than with other iron formulations, although the underlying mechanism remains unclear [25]. Although recent studies have evaluated ferric derisomaltose for lower rates of hypophosphatemia (8% versus 73–75% with FCM) [26], these findings are primarily derived from non-pregnant populations. To date, only one RCT investigated the use of ferric derisomaltose during pregnancy [27], and most available data pertain to the postpartum setting, where no significant differences in fatigue or postpartum depression were found when compared to FCM [28]. Notably, clinically relevant hypophosphatemia appears more common in patients receiving repeated or high-dose FCM regimens over extended periods, which is not typical in pregnancy-related iron treatment protocols [29].

In our study, maternal serum phosphate was not measured after FCM infusion. However, the observed improvement in maternal symptoms and the use of single-dose regimens (≤1000 mg) suggest a low likelihood of clinically relevant hypophosphatemia in this cohort.

Future research should aim to directly compare different IV iron formulations during pregnancy, particularly in terms of their biochemical safety and patient-reported outcomes, to better inform clinical decision-making in the setting of moderate to severe IDA.

## 6. Strength and Limitations

To our knowledge, no previous studies have evaluated the effects of FCM infusion on fetal wellbeing using ultrasound and cCTG analysis. The use of cCTG allowed for an objective assessment of fetal wellbeing using quantitative parameters (STV). Another strength is that fetal monitoring was performed both during iron infusion and again within 120 min, providing a dynamic evaluation of fetal response.

Nevertheless, our study has some limitations. First, as a multicenter retrospective study, there may be a selection bias toward more severe or complex cases, potentially limiting generalizability to broader obstetric populations. Data completeness may additionally vary depending on institutional practices. We attempted to mitigate this by excluding patients with incomplete outcome data, as described in Section 2.

Secondly, the absence of a placebo or control group—such as patients receiving saline solution—is another limitation. However, given that iron supplementation is the standard of care for IDA in pregnancy, a no-treatment arm would not have been ethically feasible. Therefore, our findings reflect real-world clinical practice, but do not allow for direct comparisons with untreated patients.

Lastly, our study did not assess correlations between Hb value improvement and perinatal outcomes, such as birth weight, delivery complications or neonatal wellbeing. Future prospective studies are needed to explore whether maternal anemia resolution translates into improved obstetric and neonatal outcomes.

## 7. Conclusions

Our findings suggest that FCM appears to be a safe and effective option for treating IDA during pregnancy, with no fetal adverse effects and minimal maternal side effects detected in this retrospective cohort. These results are consistent with current guidelines, which do not recommend routine fetal monitoring during FCM infusion. Nevertheless, the retrospective design, the absence of a comparator group and the lack of long-term follow-up data limit the strength of these conclusions. Further prospective studies, including head-to-head comparisons with other IV iron formulations and the evaluations of perinatal outcomes, are needed to confirm these findings and guide optimal treatment strategies.

## Figures and Tables

**Figure 1 nutrients-17-02670-f001:**
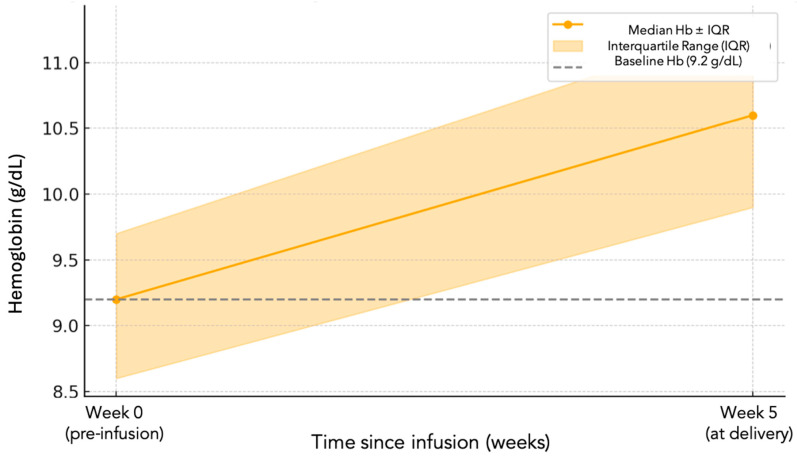
Hemoglobin levels before FCM infusion and at delivery. Median hemoglobin levels (g/dL) before intravenous iron infusion (Week 0) and at delivery (Week 5) in the total study population (*n* = 472). Error bars represent the interquartile range (IQR). Data are presented as median values with IQR.

**Table 1 nutrients-17-02670-t001:** Fetal wellbeing data during and after intravenous iron therapy and maternal side effects.

Variable	N = 472
Fetal wellbeing assessment	
cCTG analysis evaluation, *n* (%)	377 (80)
- cCTG analysis during iron therapy, *n* (%)	37 (9.8)
- cCTG analysis within 30 min after iron therapy, *n* (%)	255 (67.6)
- cCTG analysis within 30–60 min after iron therapy, *n* (%)	61 (16)
- cCTG analysis within 60–120 min after iron therapy, *n* (%)	24 (6.4)
STV value (ms)	10.0 (7.0–11.3)
Dawes and Redman criteria met, *n* (%)	371 (98.4)
Lack of high variability episodes, *n* (%)	6 (1.6)
Decelerations, tachycardia, bradycardia episodes, *n* (%)	0 (0)
US examination, *n* (%)	95 (20)
- Fetal heart rate abnormalities, *n* (%)	0 (0)
Maternal side effects	
- Lower extremity hypoesthesia, *n* (%)	1 (0.2)
- Skin rash, *n* (%)	1 (0.2)
- Hypotension, *n* (%)	2 (0.4)
- Nausea and vomiting, *n* (%)	2 (0.4)
- Skin tattoo, *n* (%)	1 (0.2)

Data are expressed as medians and interquartile ranges or absolute and relative frequencies.

**Table 2 nutrients-17-02670-t002:** Demographic and clinical characteristics of the study population and subgroups.

Variable	Total Study Population N = 472	Pre-Treatment Hb < 8 N = 53	Pre-Treatment Hb ≥ 8 N = 419	*p*-Value
Maternal age, (years)	35 (30–39)	34 (29–39.5)	35 (30–39)	0.22
BMI (kg/m^2^)	22.6 (19.9–25.2)	22.4 (19.6–25.0)	22.6 (20.0–25.2)	0.76
Nulliparous, *n* (%)	208 (44)	21 (39)	187 (45)	0.46
Twin pregnancies, *n* (%)	43 (9)	8 (15)	35 (8)	0.36
Comorbidity predisposing to IDA, *n* (%)	67 (14)	8 (15)	59 (14)	0.46
Gastrointestinal or autoimmune disorders, *n* (%)	13 (3)	0 (0)	13 (3)	0.38
Previous gastrointestinal surgery, *n* (%)	19 (4)	3 (6)	16 (4)	0.81
Gynecological disorders, *n* (%)	35 (7)	3 (6)	32 (8)	0.74

Data are expressed as medians and interquartile ranges or absolute and relative frequencies. Chi-square and Mann–Whitney U tests were performed for comparisons, as appropriate. Gastrointestinal or autoimmune disorders include celiac disease, inflammatory bowel disease and chronic gastritis. Previous gastrointestinal surgery includes bariatric surgery and bowel resection. Gynecological disorders include endometriosis, adenomyosis and uterine fibroids. BMI: body mass index; IDA: iron deficiency anemia.

**Table 3 nutrients-17-02670-t003:** Treatment effectiveness data in the total study population and subgroups.

Variable	Total Study Population N = 472	Hb < 8 Before Treatment N = 53	Hb ≥ 8 Before Treatment N = 419	*p*-Value
Hb level before IV therapy (g/dL)	9.2 (8.6–9.7)	7.6 (7.3–7.8)	9.3 (8.8–9.7)	<0.001
Ferritin level before IV therapy (ng/mL)	13.7 (7.9–16.7)	14.1 (9.7–19.1)	13.60 (7.7–16.5)	0.53
Hb level at delivery (g/dL)	10.6 (9.9–11.3)	10.4 (9.2–11.2)	10.60 (9.9–11.4)	0.13
Increase in Hb level after treatment (g/dL)	1.4 (0.7–2.2)	2.8 (1.7–3.6)	1.3 (0.6–2.0)	<0.001
Gestational age at iron IV therapy (weeks)	34.0 (29.0–37.0)	32.0 (28.0–37.0)	35.0 (29.0–37.0)	0.13
Gestational age at delivery (weeks)	39.0 (37.0–40.0)	38.0 (38.0–39.0)	39.0 (37.0–40.0)	0.89
IV iron doses (number)	1.6 ± 0.7	1.9 ± 1.0	1.6 ± 0.7	0.02
Oral iron supplementation before IV therapy, *n* (%)	308 (65)	36 (68)	272 (65)	0.90
Non-anemic patients at delivery after treatment (Hb ≥ 11 g/dL), *n* (%)	190 (40)	11 (21)	179 (43)	0.84
Hb < 8 g/dL anemia at delivery after treatment, *n* (%)	3 (0.6)	1 (2)	2 (0.5)	0.03

Data are expressed as medians and interquartile ranges or as absolute and relative frequencies. Chi-square and Mann–Whitney U tests were performed for comparisons, as appropriate. IV: intravenous; Hb: hemoglobin.

## Data Availability

The data analyzed in this study were fully anonymized. Due to ethical and institutional policy restrictions, they are available from the corresponding author upon reasonable request.

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
