# Peer review of "Intravenous Ferric Carboxymaltose for the Treatment of Iron Deficiency Anemia During Pregnancy: Effects on Maternal and Fetal Wellbeing—A Multicenter Retrospective Observational Study"

_nutrients, 2025, doi:10.3390/nu17162670_

Round 1

Reviewer 1 Report

Comments and Suggestions for Authors

Reviewer Comments to the Authors

This manuscript presents a retrospective multicentre analysis of 472 pregnant women with moderate to severe iron deficiency anaemia (IDA) treated with intravenous ferric carboxymaltose (FCM), aiming to assess both maternal and fetal safety and effectiveness. The study addresses an important clinical question and contributes valuable real-world data, particularly with the use of computerized cardiotocography (cCTG) as an objective fetal monitoring tool. Strengths of the manuscript include objective assessment of fetal well-being, comprehensive presentation of results, and alignment with current clinical practice and literature. The writing is generally clear, and references are recent and relevant. However, several major issues related to structure, clarity, consistency, and grammar need to be addressed.

Title: The title is informative and appropriately reflects the content of the manuscript. As a suggestion, the authors should consider specifying that this is a retrospective study to better set reader expectations.

Abstract: The abstract correctly presents the aim, methods, and main findings. Language, especially in conclusions, should be more cautious, avoiding strong causal claims given the observational design.

Introduction: The introduction provides a broad overview of IDA in pregnancy, its prevalence, consequences, and rationale for intravenous iron treatment. The authors cite recent guidelines and studies, which add relevance and currency to the literature cited (e.g., WHO, ACOG, and EMA). Also, the research aim is clearly stated at the end (lines 78–81), aligning well with the rest of the section.

However, there is a lack of focus and structure, it presents too many facts in quick succession, making it feel dense and unfocused. It would benefit from clearer paragraph structure, e.g.:

  1. One paragraph on the clinical problem and prevalence
  2. One on risks of untreated IDA
  3. One on treatment strategies and knowledge gaps
  4. One stating the study rationale and aims

As a suggestion, the authors should reorganise the introduction into 3–4 well-structured paragraphs, each with a clear topic sentence and progression of ideas.

Line 54: “...with rates of prenatal ID being reported as high as 19.2% [6].” This lacks geographic context. Is this a global average? From a specific region? Clarify whether the 19.2% estimate is global, regional, or based on a specific population (e.g., US, Europe).

Lines 74–77: “...a recent guideline from hematologists advising against it [13]. On the other hand, two cases of fetal bradycardia... [14]”. This brief contrast between guidelines and case reports is interesting, but the discussion is superficial. Expanding this section by explaining why guidelines do not recommend monitoring and what evidence (or lack thereof) supports this position would benefit the introduction. Also, qualify the weight of case reports versus large datasets.

Line 78–81: “...to assess fetal wellbeing through ultrasound and cardiotocographic parameters during IV iron administration...”. The aim is stated clearly but could be more compelling if framed in terms of filling a knowledge gap. Suggestion: Add a sentence before the aim to explicitly state the gap in the literature or the controversy this study addresses. For example: “Despite the increasing use of FCM in pregnancy, limited data are available on its real-world impact on fetal cardiac parameters and maternal safety, particularly when assessed with objective measures such as cCTG.”

Methods: The authors provide clear inclusion and exclusion criteria, including clinical thresholds for haemoglobin and ferritin (lines 88–92), which are appropriate for the population and research question. The study setting is clearly stated (Careggi University Hospital and Vittore Buzzi Hospital), and the period of data collection (2019–2025) is identified early (line 83). Additionally, the description of fetal monitoring methods (cCTG vs. ultrasound) based on gestational age is a valuable strength of the study design (lines 107–111).

In the following lines, I depict some areas of improvement and suggestions:

  1. Ethical considerations need more detail

Lines 85–87: “Institutional Review Board approval was not required due to the retrospective collection of fully anonymized data.”  The justification for exemption from IRB approval is brief and lacks context. It is unclear whether this is consistent with national or institutional ethical guidelines.

The authors should cite the specific national or institutional policy that permits IRB exemption in retrospective studies using anonymised data. They could add a sentence such as: “This is in accordance with [Institutional or National Guidelines], which waive ethical review for retrospective analyses involving anonymised data.”

  1. Missing information on informed consent

Later in the manuscript (line 318) they state: “Informed consent was obtained from all subjects…”. If informed consent was indeed obtained, it should be explicitly stated in the Methods section where data collection and ethical considerations are first introduced. Otherwise, it creates confusion about whether the study was fully anonymised or based on identifiable patient data.

The authors should move or restate the informed consent clarification in the Methods section. Clarify whether this was written consent for treatment only, or also for data usage in research.

  1. Clarification needed in some procedures

Lines 112–116: “The single administration did not exceed 20 mg/kg of body weight and the maximum cumulative weekly dose was 1000 mg of iron (2 administration a week)…” The sentence has unclear grammar and is difficult to follow. For example, “2 administration a week” should be “2 administrations per week”. Rewrite for clarity: “Each administration did not exceed 20 mg/kg of body weight, and the maximum cumulative weekly dose was 1000 mg, typically divided into two infusions per week.”

  1. Handling of missing data is not addressed

There is no mention of how missing data were handled—particularly relevant in retrospective studies where not all records are complete (e.g., patients without cCTG or Hb follow-up).

Add a brief statement such as: “Patients with incomplete outcome data were excluded from the corresponding analyses. No imputation was performed.”

  1. Statistical tests require justification

Lines 117–127: The authors used Wilcoxon signed-rank and Mann-Whitney U tests but do not clearly explain the rationale for their use. It is only briefly noted that data were tested for normality using Shapiro-Wilk, but they do not report whether data were normally distributed or not. Moreover, there is no detail about the sample sizes in the tested subgroups, which affects the suitability of non-parametric tests.

Suggestion: Add “As Hb and ferritin values were not normally distributed (p < 0.05, Shapiro-Wilk test), non-parametric tests were used.” If the data were normally distributed and they still used non-parametric tests, the authors should justify this choice.

  1. Use of software could be expanded

Line 126: “All analyses were performed using SPSS software (version 30.0.0.0, IBM Corp., Armonk, NY, USA)”. Mention whether two-sided p-values were used and whether any corrections for multiple comparisons were applied. If not relevant, clarify that all tests were two-tailed and no adjustments were made.

Additional comment: the authors may improve transparency by:

  • Including a STROBE checklist in supplementary materials (standard for observational studies).
  • Reporting baseline characteristics by outcome subgroup (e.g., those with/without Hb response).

Results: The results are logically structured, progressing from baseline characteristics to fetal monitoring findings, maternal outcomes, and haematological responses. This makes the section easy to follow.

The use of multiple tables provides a clear overview of patient demographics (Table 1), haematological changes (Table 2), and maternal and fetal safety outcomes (Table 3). The inclusion of a subgroup analysis based on baseline haemoglobin (<8 g/dL vs ≥8 g/dL) adds analytical depth and clinical relevance (lines 162–166).

However, some issues must be addressed:

  1. Inconsistent reporting of maternal side effects
  • Line 38 (Abstract): “Maternal side effects... occurred in 4 patients (0.8%).”
  • Line 153 (Results): “Maternal side effects were recorded only in 7/472 (1.4%) cases...”

There is a discrepancy in the number of maternal adverse events reported between the abstract and the main text. Ensure consistency across sections or clearly differentiate between types of events (e.g., minor vs serious, or immediate vs follow-up events). State the correct number and revise both locations accordingly.

  1. Ambiguity in Table 1 (baseline characteristics)

Line 136–137: “Twin pregnancies n (%) 43 (9)”. This is likely a formatting or typographical error. As written, it suggests that 43% of pregnancies were twin pregnancies (which is implausible), while the number in parentheses suggests 9 twin pregnancies. Correct to: “Twin pregnancies, n (%): 43 (9)”. More generally, ensure clear alignment and labelling of variables and units across all tables.

  1. Figure 1 needs better axis labelling

Line 160: “Figure 1. Mean hemoglobin level improvement...” The figure’s x-axis and y-axis labels are vague. There is no indication of time points or number of patients per time point, and the caption lacks detail (e.g., SD bars, stratification by subgroup). Add axis labels such as “Time since infusion (weeks)” and “Haemoglobin (g/dL)”. Clarify if the figure represents mean values for the entire sample or by subgroup (Hb <8 vs ≥8). Include error bars and state in the figure legend whether they represent SD or IQR.

  1. Lack of raw numbers in some outcomes

Example (Line 167): “190 (40%) achieved a complete resolution of anemia…” The percentage is given, but the denominator and the definition of resolution (Hb ≥11 g/dL) should be reiterated. Similarly, the number of patients lost to follow-up (if any) is not reported. Restate the denominator for clarity: “190 out of 472 patients (40%)…” Specify how many were evaluated at delivery for this outcome and whether any were excluded due to missing Hb data.

  1. Minor formatting and clarity issues

Percentages and sample sizes should be reported consistently in the same order: e.g., “n (%)” rather than mixing styles (e.g., “65 (308)” vs “308 (65%)”). Standardize decimal precision across tables. For instance, Hb values are reported with inconsistent decimals (e.g., “9.1” vs “10.7 ± 1.22”).

Discussion: The discussion section adequately contextualises the findings, linking them to relevant literature and current clinical guidelines (e.g., lines 186–213, 214–220). The authors integrate recent pharmacovigilance data and RCT evidence (lines 196–202, 257–258), which strengthens the relevance of their conclusions. The attention to clinical nuances such as maternal comorbidities, oral iron absorption mechanisms, and iron pharmacodynamics (lines 232–246) shows a commendable depth of knowledge. The acknowledgement of the lack of phosphate monitoring and retrospective design (lines 285–290) adds credibility and transparency.

However, there are important concerns that authors must revise:

  1. Overstatement of conclusions

Lines 189–190: “...our results showed a reassuring safety profile for the fetus, with no adverse events or cardiac rhythm alterations...” Given the retrospective, observational nature of the study and the lack of a control group, the term “reassuring” might overstate the certainty of foetal safety.

Suggestion: rephrase to reflect the observational limitations, e.g.: “No fetal cardiac rhythm alterations were observed in this cohort, supporting the overall safety profile of FCM. However, the retrospective design limits the ability to draw causal inferences.”

  1. Uncritical presentation of guideline alignment

Lines 296–298: “Our study corroborate the indication of recent guideline to not monitor fetal heart reate during or after IV iron infusion.” This conclusion is too strong given the limited observational design and the absence of a comparator group that was monitored vs. not monitored. Reword to reflect the supporting—but not confirming—nature of the findings: “Our results are consistent with recent guidelines suggesting that routine fetal monitoring may not be necessary during FCM infusion, though further prospective studies are warranted.”

  1. Comparison to other iron formulations

Lines 270–280: The discussion of ferric derisomaltose and hypophosphataemia is scientifically interesting and relevant. However, this comparison risks shifting the focus of the discussion too heavily onto a different product that was not evaluated in the study. It also includes speculative language and data from non-pregnant populations. Reframe this discussion as part of the “future research” or “clinical perspective” angle, and clearly note that the study did not assess other iron formulations: “Although our study focused on FCM, future research may compare it directly with other IV formulations, such as ferric derisomaltose, particularly in light of emerging evidence on differential rates of hypophosphataemia.”

  1. Absence of discussion on potential biases

The discussion does not critically assess the potential for selection bias, data completeness, or centre-level variability, despite being a multicentre retrospective study. The authors must add a paragraph on potential biases and generalisability: “As this was a retrospective study based on two referral centres, there may be selection bias toward more severe or complex cases. Additionally, data completeness may vary depending on institutional practices, which could influence outcome ascertainment.”

  1. Clarification of Hb outcomes and clinical relevance

While increases in Hb are reported (lines 248–256), the discussion could do more to reflect on clinical thresholds and what constitutes a meaningful response, especially in relation to maternal or perinatal outcomes. Including interpretation of the response magnitude is required: “The observed average increase of 1.6 g/dL is clinically meaningful, particularly as 40% of patients achieved Hb levels above 11 g/dL at term. However, without correlating to perinatal outcomes such as birth weight or delivery complications, the functional impact remains uncertain.”

Conclusion: The conclusion effectively summarises the main findings of the study, notably the increase in haemoglobin levels and the absence of observed adverse foetal effects following FCM treatment. The recommendation of FCM as a first-line intravenous treatment for IDA in pregnancy is supported by the data presented and aligns with current clinical practice and guidelines.

But, again, strong claims should be tempered. In lines 296-298, the authors state ““FCM is a safe and effective treatment in pregnancy, without significant side effects either on mother and fetus. Our study corroborate the indication of recent guideline to not monitor fetal heart reate…” This wording implies causality and generalisability that go beyond what a retrospective observational study can support, especially in the absence of a control group or randomisation. Rephrase to reflect the observational nature of the study: “Our findings suggest that FCM appears to be a safe and effective option for treating IDA during pregnancy, with no observed fetal adverse effects and minimal maternal side effects in this cohort. These results are consistent with current guidelines, which do not recommend routine fetal monitoring during FCM infusion.”

Additionally, a brief sentence acknowledging key limitations is strongly recommended: “Nevertheless, the retrospective design, absence of a comparator group, and lack of long-term follow-up data limit the strength of these conclusions.”

The conclusion also misses the opportunity to point toward future research needs, particularly comparative trials of FCM versus other iron formulations or studies evaluating perinatal outcomes. The authors may conclude with a forward-looking sentence: “Further prospective studies, including head-to-head comparisons with other intravenous iron formulations and evaluations of perinatal outcomes, are needed to confirm these findings and guide optimal treatment strategies.”

References: The references are up to date and largely relevant to the topic. Just some formatting inconsistencies need to be addressed.

Language, grammar, and formatting style

There are frequent grammatical issues throughout the manuscript, including incorrect article usage, verb tense errors, and typographical mistakes. English language edit is strongly recommended.

Tables require cohesive formatting with greater attention to detail and adherence to academic standards.

Comments on the Quality of English Language

There are frequent grammatical issues throughout the manuscript, including incorrect article usage, verb tense errors, and typographical mistakes. English language edit is strongly recommended.

Reviewer 2 Report

Comments and Suggestions for Authors

It is an interesting study where authors evaluated impact of IV ferric carboxymaltose on maternal and fetal wellbeing. However, due to nature of one-arm design, the results should be interpreted with caution. And the following technical issues should be addressed further if a revision would be invited.

  1. The methods are too simple and poor organized. There are more information needed. For example, about treatment, authors did not provide the duration of treatment. Also they did not provide clearly outcomes of interest and when to collect them.
  2. Authors said in line 125 that “A non-parametric Mann-Whitney U test was used to compare the Hb increase between these two subgroups.”, but they use mean and sd to describe the data. Generally, if you use A non-parametric Mann-Whitney U test, median and IQR are better to describe.
  3. It is poor for organization of results. Subtitle may be needed to distinguish the results for mothers and fetals. It seems not to present clearly result of fetal wellbeing. It is better to show results before and after treatment together, not just increase, but before, after and increases.
  4. It is unclear what other treatment such as dietary intervention are used for these patients.

Round 2

Reviewer 1 Report

Comments and Suggestions for Authors

The manuscript has been improved after the revisions made.

Reviewer 2 Report

Comments and Suggestions for Authors

Authors have addressed my concerns, thanks.